# New Methodologies Indicating Adhesive Wear in Load Step Tests on the Translatory Oscillation Tribometer

**Gregor Patzer** [1,*] **and Mathias Woydt** [2,*]

1   Optimol Instruments GmbH, 81369 Munich, Germany
2   Materials Tribology Lubrication, 12203 Berlin, Germany
*   Correspondence: gregor.patzer@optimol-instruments.de (G.P.); m.woydt@matrilub.de (M.W.)

**Abstract:** When looking in detail at analyses of the tribological load-carrying capacity of lubricants, it becomes apparent that an exclusive evaluation of the evolution of the coefficient of friction alone cannot provide any sufficient criteria for determining the occurrence of adhesive failure. For this reason, extending the knowledge base by combining several criteria in order to draw a clearer picture of adhesive wear mechanisms is urgently required. This can be achieved by combining the evolution of coefficient of friction with stroke signals and/or the electrical contact resistance and/or contact temperature and/or acoustic emission and/or stroke zero position, frictional power input and further derived parameters.

**Keywords:** SRV; seizure; scuffing; zero stroke position; electrical contact resistance; acoustic emissions





## 1. Introduction

Once a tribosystem is underway to scuff, seize, gall or score, the failure can occur quickly. The aforementioned synonyms are summarised under the wear mechanism of adhesive wear. This may occur in any machinery with tribosystems either running dry (unlubricated) or are lubricated by greases or oils. The loss of protection against adhesive wear of lubricants over time or under peak loads initiates such failures. The pass or OK load or non-seizure load of lubricants can be established in tribometers. Any weld load or stoppage of the test machine represents the ultimate and clear end of the test due to adhesive wear. The question is, if the loss in protection against adhesive wear mechanisms occurred spontaneously (cliff), or is there a transition zone from safe operation to scuffing, e.g., grey zone, as a function of Hertzian contact stress, velocity and temperature? Therefore, finding a reliable and competent early warning methodology is required, especially in the development of lubricant formulations, but also for the condition monitoring of used oils.

The tribological load-carrying capacity of lubricants is defined on the translatory oscillation tribometer (SRV; abbreviation from German: Schwingung, Reibung, Verschleiss) by load step tests under the regime of mixed/boundary lubrication with standardised processes and parameters according to ASTM standards D5706 [1] and D7421 [2] or as ISO 19291:2016-12 homologues. This includes the increase in normal force in constant increments until the lubricant cannot prevent the movement of friction components and adhesive failure occurs or the tribometer specific maximum load of generally 2000 N (2500 N) is reached. The functional and tribological capabilities of base oils, additives and novel concepts have greatly improved, and formulated lubricants are bringing test methodologies to their limits.

Through the latest high-performance electronics in connection with appropriate software functionality, it has now become possible to deal with big data volumes and to analyse signals at high resolutions. Besides conversion of measuring values in derived result channels such as the effective value of the coefficient of friction or friction energy, it is possible to integrate semi-automatic software algorithms that already analyse incoming measuring data with regard to distinctive features.

The present article shows ideas and discusses approaches on how the interpretation of load step tests beyond the standardised resulting parameters can lead to safer identification of adhesion, which is a loss of functionality of the lubricant and enables deeper insights into its origin.

## 2. Experimental Procedure

Within the scope of the test methods, lubricants, which served as samples tested in international SRV round robin tests, were subject to the tribological load-carrying capacity test according to ASTM D5706 for greases and/or ASTM D7421 for oils. These lubricated tests ran in the regime of mixed/boundary lubrication. During testing, besides the usual operating conditions and measurements, such as normal force, temperature, frequency, stroke and coefficient of friction, further quantities were recorded for interpretation: electric contact resistance, acoustic emission, power consumption of drive unit, zero stroke position of oscillation, effective value of coefficient of friction ($cof_{eff.}$) and as high-speed data (position of upper specimen, coefficient of friction, sliding speed, electrical contact resistance and acoustic emission). Furthermore, frictional energy, friction power as well as crest factor are optionally determined through collected measuring data.

### 2.1. SRV Tribometer

The tests are carried out on SRV of the fifth generation (Figure 1). This differs from its predecessor models by a changed electronic concept (Field Programmable Gate Array system, FPGA) with the latest user software and further mechanical developments. However, the SRV basic principles, which are binding for compliance with all relevant standards, still apply: generation of motion through the electromagnetic linear motor, applying normal force through a gear shaft spring system, conversion of friction force into the coefficient of friction directly in the amplifier electronics, acquisition of friction force through piezoelectric sensors (Figure 2). The FPGA system allows many parallel data operations as well as a flexible parametrisation and adjustment of electronic behavior through well-engineered firmware concepts. Thus, time-critical secondary resulting parameters are calculated directly in the hardware without time falsification and are recorded as measuring channels in the evaluation software.

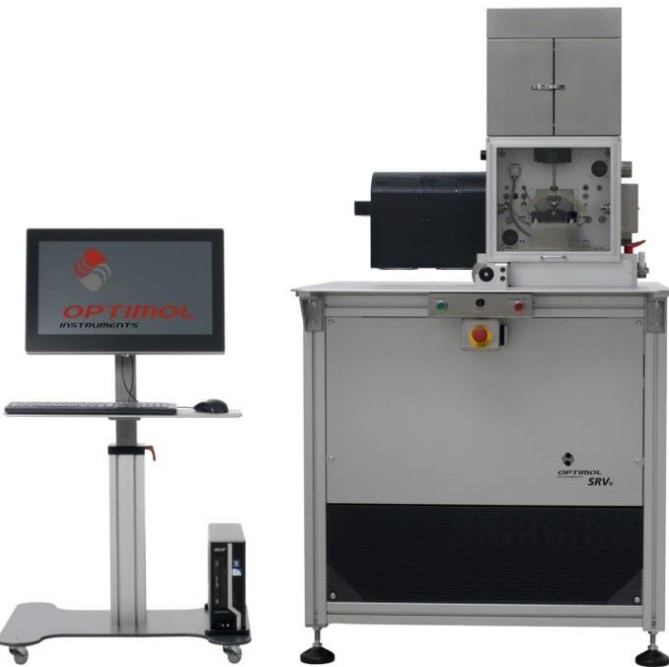

**Figure 1.** SRV model 5 tribometer with PC desk.

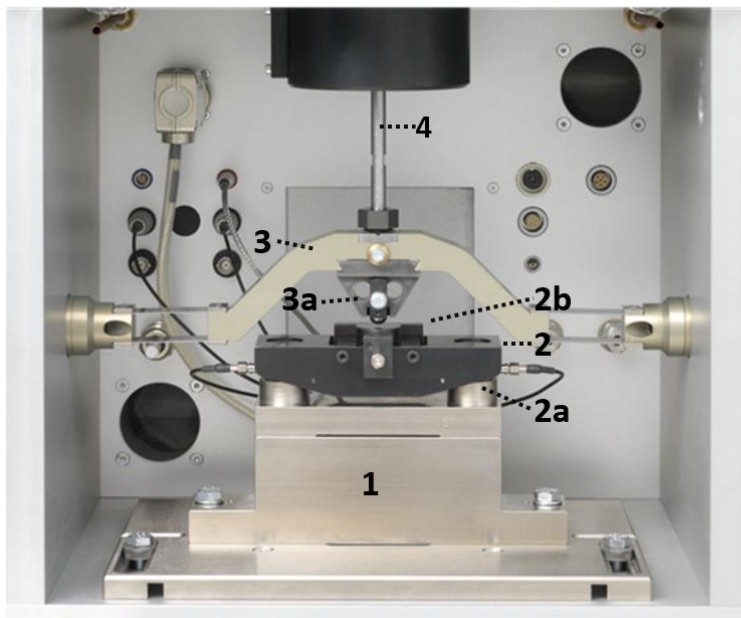

**Figure 2.** Overview of the most important elements within the SRV test chamber.

The SRV machine has a specific drive unit. The sinusoidal movement of the linear actuators is controlled by a displacement sensor and not strictly enforced by an excentre or a crankshaft. This implies that depending on the speed of the control unit in relation to the spontaneity of the adhesive event and in view of the actual level of friction force, the stroke may shut, vary and/or break in.

Such events may occur during testing of the extreme pressure or of friction and wear properties. As the movement is not enforced by a crankshaft, the SRV actuator does not interfere by the design of the tribometer with tribochemistry and tribo-oxidation in the tribocontact.

An increase in friction results in a reduction in the stroke as the actuator moves the sample at the time of the adhesive event with a defined force. The control unit increases the force (by power) in order to assure the predefined stroke. If the adhesive event disappears and the coefficient of friction reduces, the stroke will exceed the pre-set value unless the control unit reduces the force. Sharp changes in the friction force also distort the sinusoidal movement.

In consequence, a sharp increase in friction force coincides with fluctuations of the stroke. Tentative events of adhesive failure are displayed by two quantities: the friction force or the coefficient of friction and in parallel by the stroke length and zero stroke position.

Due to this specific actuation design, the current or power consumed by the oscillation motor can be used as an additional criterion to assess adhesive events in the tribocontact. In consequence, the friction signal can be combined with stroke length and zero stroke position and/or the power consumption of the actuator.

### 2.2. Load Step Tests according to ASTM D5706 and D7421

Both standard tests specify parameter progressions for load step tests on the SRV with ASTM D5706 for greases and ASTM D7421 for oils. The parameter progressions for both standards are mostly identical, and the only difference is the amplitude with which the upper specimen is moved over the lower specimen. The temperature is generally selected between 50 °C and 150 °C in 10 K steps each and kept constant. The operating conditions for extreme pressure (EP) tests are shown in Table 1.

**Table 1.** Parameter sets for SRV load step tests according to ASTM D7421 and D5706.

| Quantity | ASTM D7421 | ASTM D5706, Procedure B |
|---|---|---|
| Test load | Running-in under 50 N for 30 s, then 100 N for 15 min followed every 2 min by an increase of 100 N until maximum load is reached or adhesive failure | Running-in under 50 N for 30 s, then 100 N for 15 min followed every 2 min by an increase of 100 N until maximum load is reached or adhesive failure |
| Frequency | 50 Hz | 50 Hz |
| Stroke | 2.0 mm | 1.5 mm |
| Temperature | Typical: 50 °C, 80 °C and 120 °C (2 repeats) | Typical: 50 °C and 80 °C (2 repeats) |
| Test period | max. 55 min | max. 55 min |
| Lubricant volume | 0.3 mL | Grease caliper of 1 mm in height |

A steel ball (diameter 10 mm) as an upper specimen as well as a disk (diameter 24 mm; height 7.9 mm) as a lower specimen serves as test pieces. Both are made of quenched and tempered ball bearing steel 100Cr6 (AISI 52100) according to the specifications of ASTM D5706 and D7421.

Thus, during testing (Figure 3), the lubricant is subjected to an initial mean Hertzian contact pressure $P_{0mean}$ of approximately 1.8 GPa ($F_N$ = 200 N) to 3.9 GPa ($F_N$ = 2000 N).

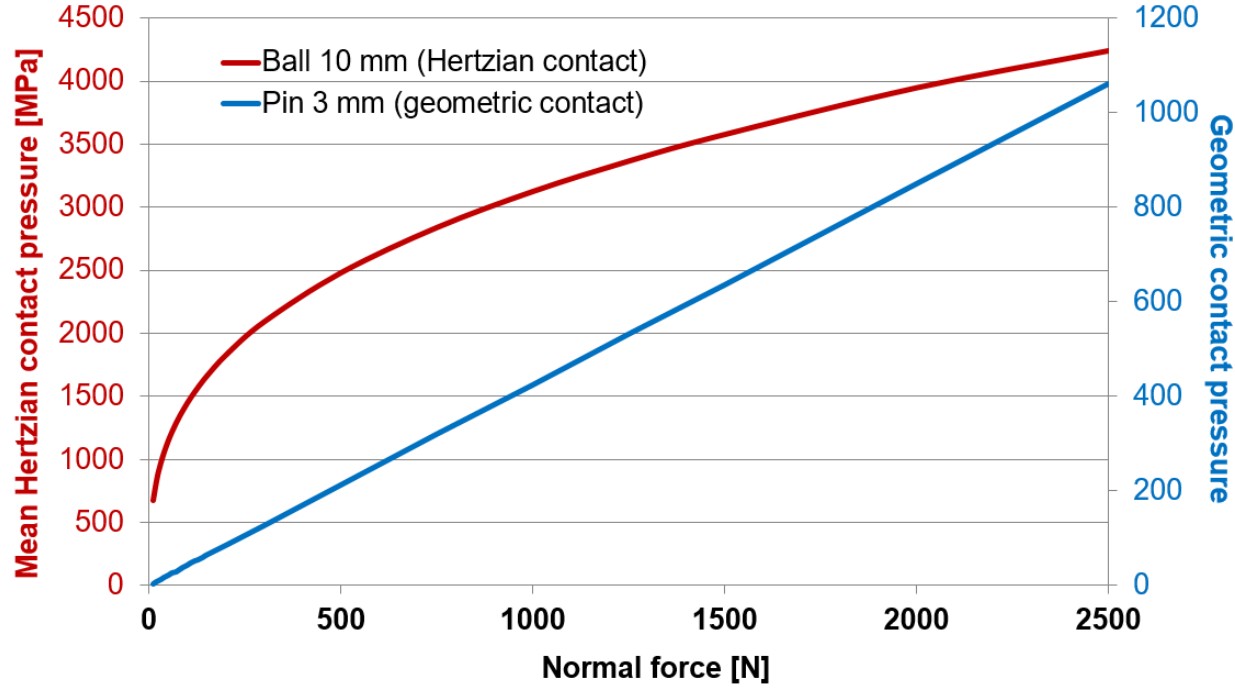

**Figure 3.** Evolution of initial and mean Hertzian contact pressure (left, $P_{0mean}$) as an example of a point contact as well as a pure flat geometric contact (right, $P_{0geom.}$).

As a result of testing according to ASTM D5706 and/or D7421, the so-called pass load is indicated. This is the last applied normal force, which could be held for 2 min without adhesive failure approaches. The running-in phase of the load step tests under $F_N$ = 100 N is 15 min long. Figure 4 represents an unambiguous case where the machine stopped due to the strong adhesive wear or welding. It is also apparent that the electrical contact resistance began steadily to decrease around three load steps before the stoppage of the machine.

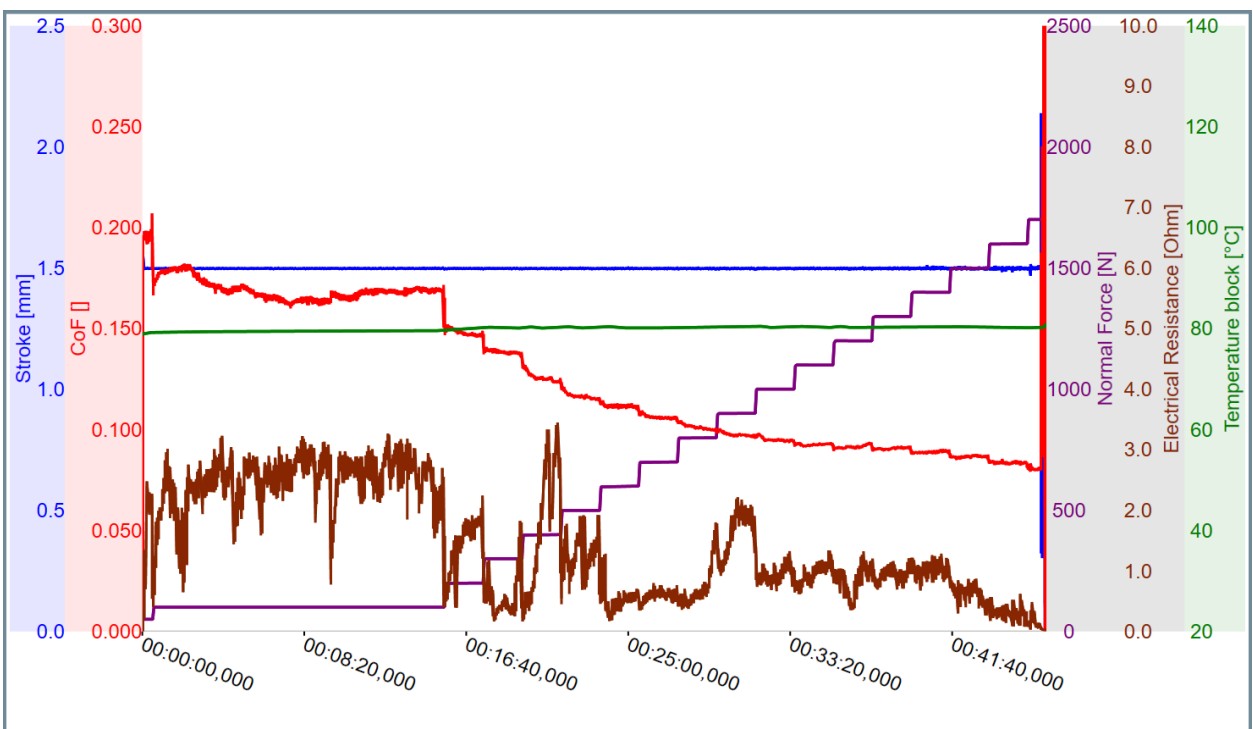

**Figure 4.** Evolution of the coefficient of friction (red), stroke (blue) and electrical contact resistance (brown) in the load step test according to ASTM D5706 with a pass load of 1600 N of the grease. The sample temperature is represented as a green line.

### 2.3. Evolution of Past Failure Criteria

Within the oil industry, galling or scuffing is understood as a form of lubricant-related adhesive wear and/or of the loss of the extreme pressure properties of lubricants. The material aspects are not perceived. The early ASTM D5706-95 covered this understanding with the following definition and criteria:

Seizure, n—localised fusion of metal between the rubbing surfaces of the test pieces;

Discussion—in this test method, seizure is indicated by a rise in the coefficient of friction, over steady-state, of greater than 0.2. In severe cases, a stoppage in the motor will occur.

As shown in Figure 4, the stoppage of the motor (i.e., actuator) is the clearest indication of scuffing. By the revision of D5706 in 2002, some clarifications in order to identify adhesive wear mechanisms were added, such as:

a. "Sharp rise" in the coefficient of friction;
b. Associated with a value "greater than 0.2 over 20 s", because seizure or scuffing is a spontaneous event and the ability of the lubricant to "heal" the contact zones, the created nascent surfaces need to be regarded. The definition in ASTM D5706-02 is identical to the one stated in DIN 51834-2:1997. ISO 19291:2016-12 uses the same definition.

The revision of DIN 51834-2:2004 detailed in a note that the level of coefficient of friction and/or spikes in the friction signal do not consequently indicate adhesive wear as follows:

a. Under dry friction, couples presenting coefficients of friction greater than 0.35 are known without signs of adhesive wear in the wear scars/tracks (metallurgical aspect);
b. Lubricants are normally dedicated to reducing friction, so the coefficient of friction should be in SRV tests below 0.2;
c. During running-in, short spikes of the coefficient of friction can be observed, not consequently indicating adhesive wear.

It must be noted that the SRV round robin tests related to solid bonded films revealed coefficients of frictions above 0.3 for unlubricated oscillating couples without signs of material transfers in the wear scars and tracks. This already proves that the set limit of 0.35 does not consequently indicate adhesive failures or a malfunction of the lubricant. Nevertheless, such a set limit represents extremely high friction for lubricants if occurring in steady-state.

The examples in the annex of ASTM D5706-05 explain the relationships that "different manifestations of the friction force curve do not necessarily need to be indicative of adhesion having occurred". Several typical evolutions of the friction signal were evaluated by the standardisation working groups with a narrative interpretation.

### 2.4. Consensus in Evaluating the Criteria

The motivation for evaluating the criteria is as follows:

a.      Streamlining of the common understanding of pass/OK load in SRV testing;
b.      Defining a meaningful set of limits for the different criteria reflecting practical experiences.

The DIN51834 working group validated jointly, during several meetings, a set of load step test diagrams. For combining the coefficient of friction with the stroke, the consensus displayed in Table 2 was established.

**Table 2.** Current consensus within DIN 51834-2:1997 regarding combining criteria for the identification of adhesive failure.

| | **Fluctuations in Friction** | **Fluctuations in Stroke** |
|---|---|---|
| **Likely no occurrence of adhesive wear mechanisms** | **Small increase**: $\Delta$CoF < 0.04 <br> -Unstable signals and small increases may be interpreted as adhesive wear but are likely related to abrasive wear mechanisms due to wear particles stuck in the tribocontact. | Unstable and ragged signal: <br> if $\Delta x < \pm 10\%$ of stroke, likely no adhesive wear mechanism. |
| **Very likely presence of adhesive wear mechanisms** | Strong increase: $\Delta$CoF > 0.10 <br> -Reaction layer was pierced, resulting in metal-to-metal contact. <br> -Short turning of the ball in the holder. | Strong fluctuations: <br> if $\Delta x > \pm 10\%$ of stroke in combination with homologue fluctuations in friction signals, adhesive mechanisms are likely. |
| **Wear mechanism** | Inspection of wear track and scar by LOM (Light Optical Microscopy) | |
| **Skepticism** | If the friction signal in the combination of the stroke signal or the wear mechanisms is difficult to assign to adhesive wear, repeat the test. | |

## 3. Additional Measurements

The evolution of friction with the associated stroke can be coupled with the electrical contact resistance and/or acoustic emission. Both are non-destructive techniques for online monitoring of tribological tests. They enable in situ investigations of the friction and wear processes in sliding and rolling contacts in real-time.

### 3.1. Measurement of Electrical Contact Resistance

Normally, all metals and ceramics are covered by a more or less thick and adherent natural layer composed primarily of oxides, but also hydroxides and/or hydrates, which prevent adhesive wear. When the subsurface region flows due to high-contact pressures or high-shear stresses, nascent metallic surfaces will be formed, which weld adhesively together unless lubricant additives form a new solid reaction layer [3]. The plastic flow of the subsurface regions disturbs any tribo-oxidatively formed oxide and reactions layers or integrity of coatings, which may protect against adhesive failures. At this stage, extreme pressure additives in greases or oils prevent adhesive failure by chemical reactions or physical film from forming. The electrical contact resistance between both specimens is essentially determined by the electrical resistance properties of the formed reaction layer or solid tribofilm.

In this respect, the measuring parameter of the electrical resistance can only be used for a certain group of lubricant types whose chemical components are not electrically conductive and will also not generate any electrically conductive reaction products on the surfaces of the friction components. In consequence, the electrical contact resistance is not a clear indicator but represents an additional indicator. If these prerequisites are fulfilled, it can be stated that:

1.  After the beginning of the test, the electrical resistance first increases, then stabilises at a certain level;
2.  For some lubricants, the electrical resistance shortly collapses after the increase in normal force and then restabilises;
3.  In most cases, the electrical contact resistance hardly reacts to changes in the normal force;
4.  The electrical resistance approaches zero just before adhesive failure (Figure 5).

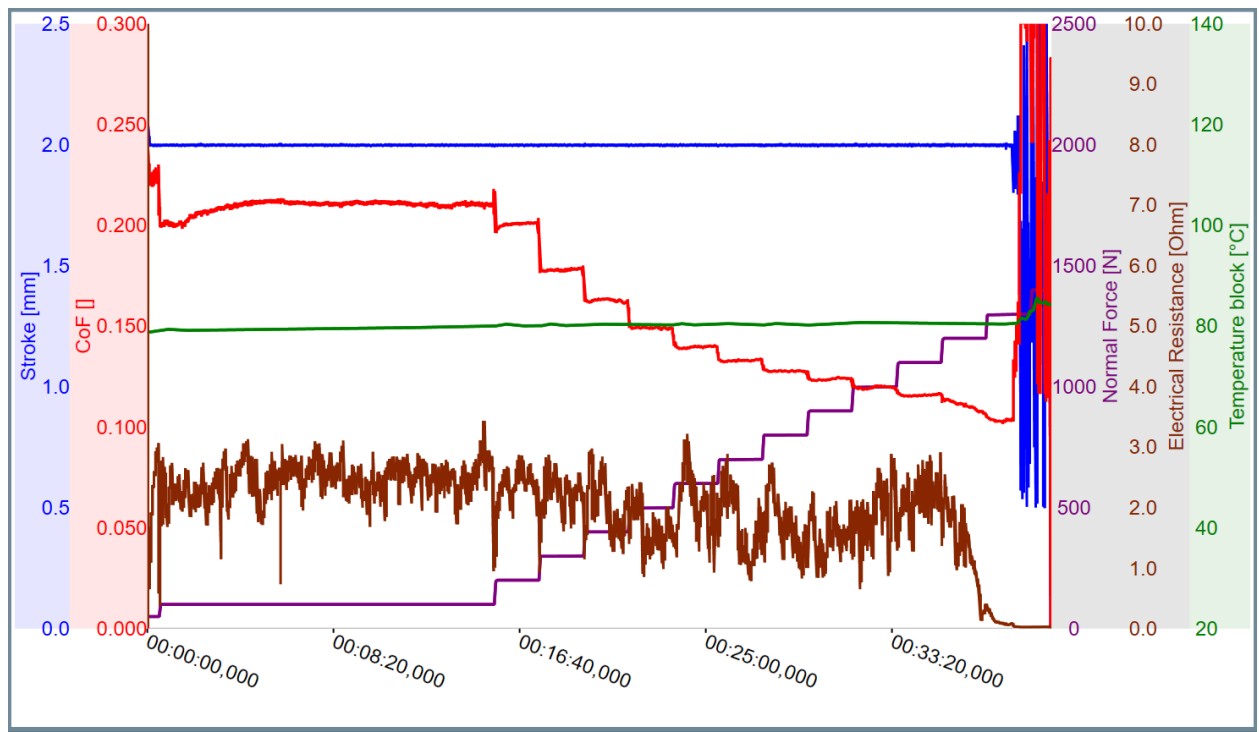

**Figure 5.** Evolution of the coefficient of friction (red), stroke (blue) and electrical contact resistance (brown) in the load step test according to ASTM D5706 with a pass load of 900 N of the grease. The green line represents the sample temperature.

Figure 4 illuminates the evolution of electrical contact resistance (brown curve) during a load step test. After each increase in normal force (purple curve), a short drop in electrical contact resistance is observed, which recovered fast. Some load steps before adhesive failure, the electrical contact resistance continuously decreased, indicating the upcoming scuffing. For the grease represented in Figure 5, changes in normal force (purple curve) hardly influence the level of the electrical resistance. The sharp drop in the electrical contact resistance at 1200 N signals up-coming premature scuffing because at this load stage, the wear of the reaction layers is faster than their rebuilding. Figure 6 illuminates an opposite behavior where, during running-in, the electrical contact resistance collapsed without any effects in the signal of the coefficient of friction and stroke. No indications of adhesive failure were shown in terms of friction and stroke because this reaction layer is of conductive nature. Figures 4 and 5 show that the stroke remained smooth over the entire test until failure occurred.

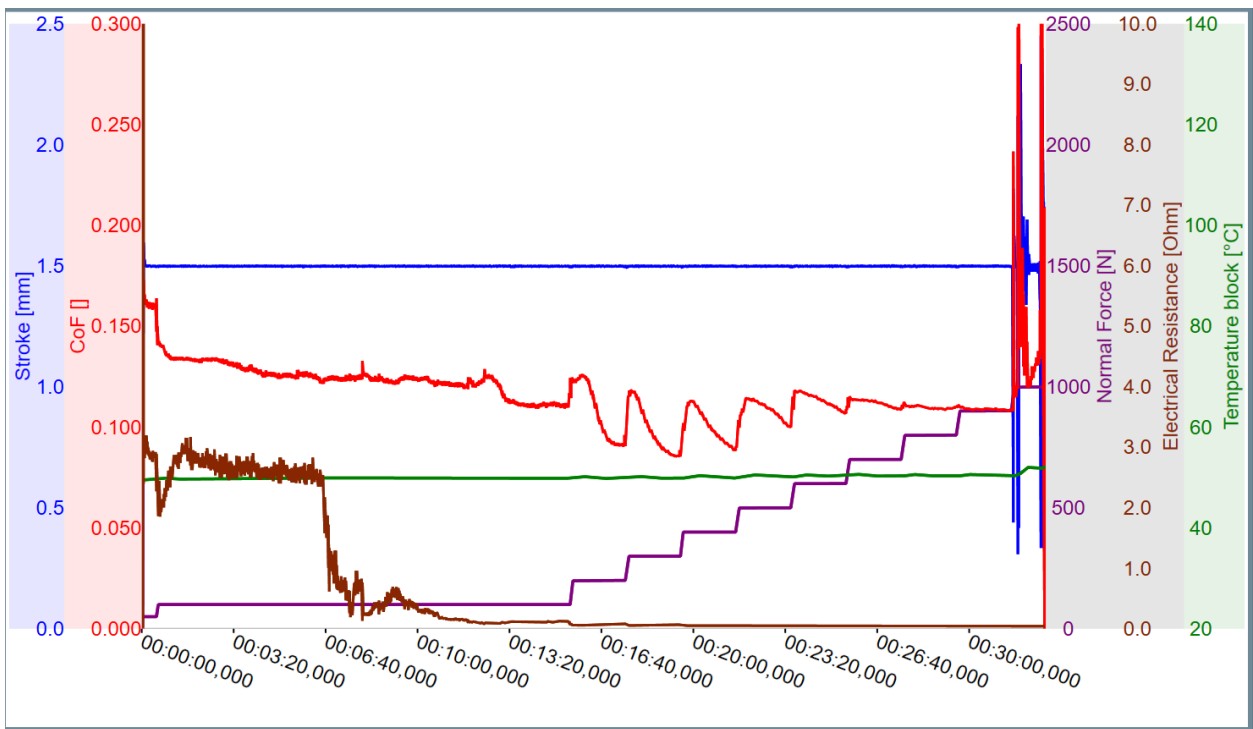

**Figure 6.** Evolution of the coefficient of friction (red), stroke (blue) and electrical contact resistance (brown) in the load step test according to ASTM D5706 with a pass load of 900 N of the grease (Sample temperature: green).

The signals of the friction in Figure 7 suddenly jumped at 1200 N on a higher steady level (top) and continued to decrease with increasing load steps. After the "cliff", the friction and stroke signals are ragged with an associated increase in activities of acoustic emissions (Figure 7, lower). The electrical contact resistance dropped sharply, and the signals of friction and stroke are slightly ragged and do not meet the criteria as in Table 2. The zero-stroke position in lower Figure 7 is increasingly oscillating early before the sharp jump ("cliff") in coefficient of friction occurs. The motor power responded with a ragged signal at 1200 N upon the adhesive event. Figure 7 displays an evolution of the test, which is interpreted in two ways:

a. Pragmatic users conclude that as long as the machine operates, no scuffing is seen, independent of how the signals are ragged and state a pass/OK load of 1900 N or 2000 N;

b. Fearful users rate any jumps and/or fluctuations in the signals as a roughening of surfaces due to adhesive wear mechanisms but agree on the second origin of abrasive wear particles formed by wear. They would report 1100 N. However, the lubricant underperformed above that point of >1100 N in terms of extreme pressure protection.

One may also conclude that, between both positions, persists a region of scuffing risks ("grey zone"). Figure 7 also shows the electrical contact resistance indicating a sudden rupture of the solid tribofilm because the contact resistance drops to zero and the solid tribofilm does not reform until the end of the test. Clearly, at 1100 N, the lubricant lost its capability to protect the surfaces under extreme pressure conditions. The width of such a grey zone depends on the operating temperature because of the kinematic and thermodynamic reactivity of additives with surfaces as a function of temperature. Recommending a formulation for a range of practical applications requires a functional EP profile determined for different temperatures.

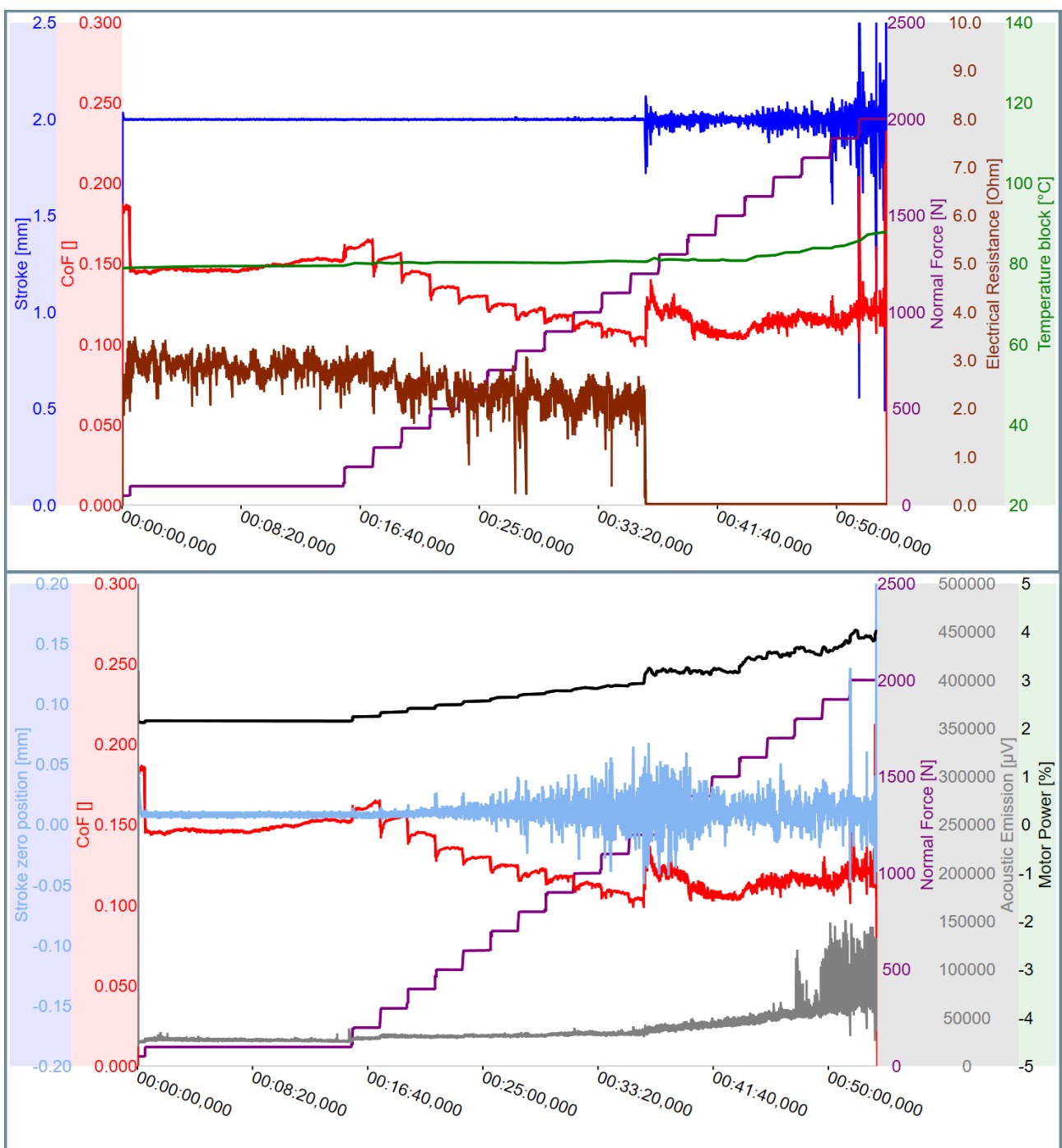

**Figure 7.** Evolution of the coefficient of friction and additional measurements in the load step test according to ASTM D5706 (top: evolution of stroke (blue) and electrical contact resistance (brown); lower: evolution of the zero-stroke position (light blue) and acoustic emission (grey)).

### 3.2. Measurement of Acoustic Emission at the Lower Disk Specimen

The formation of wear particles by abrasion and/or fatigue and/or delamination as well as the rupture of welded nascent micro-asperities, generate elastic waves propagating through the specimen [4,5]. These events on the microscale announce wear mechanisms before they become apparent at the macroscale and/or in the friction force signal. Acoustic emission represents another tool for the early detection and/or confirmation of adhesive wear (scuffing). The acoustic emission sensor is positioned at the lower disk specimen of the tribosystems. The emitted acoustic performance is a corresponding indicator for the frictional EP behavior of the tribosystem.

After running-in, the friction in Figure 8 is lowered by each load increase. As in Figure 7, the friction decreased in Figure 8 continuously with increasing load, until suddenly it jumped up at 500 N. At 500 N, the coefficient of friction with the stroke indicated a very short instability of which the origin is permanent as per the electrical contact resistance signal (brown curve). Within the test procedure, Figure 8 shows that the development of acoustic emission and stroke zero position (light blue curve) indicated an early upcoming change in the frictional EP behavior. Two to three load steps prior to the stoppage of the machine, the acoustic noise activity, and the oscillation of the stroke and zero stroke position indicated signs of changes into the wear mechanism of adhesive wear. The evolution of the power consumption of the drive unit (black) follows the load increase and changes above 600 N into a distorted shape but was not as indicative as the other measurements (compared with Figure 7, below). The electrical contact resistance displays exactly at 600 N the clear loss of solid tribofilm formation, even if the test continued until 2000 N. On the other hand, beginning between 400 and 500 N, the zero-stroke position and the acoustic emissions advertised the beginning loss of extreme pressure protection. By comparing the stroke in the lower diagram of Figure 8 with the stroke zero position at the top of Figure 8, it can be deduced that the stroke zero position is a forward indicator in view of the stroke itself.

### 3.3. Drive Power of Oscillation Motor

The peak value of the electric drive power is a direct measure for the electric power being necessary for the implementation of oscillatory motion against the friction and to assure its zero-stroke position. The value of the drive power is issued per period. The power to be applied essentially consists of two aspects:

1. Generation of the kinetic energy of the drive motor through permanent changing acceleration and deceleration of the motor during an oscillation motion;
2. Overcoming the friction forces generated during the friction contact.

Due to the mechanical structure of the actuator, the kinetic energy of the drive motor is independent of the applied normal force at the friction contact. In this respect, each increase in drive current can be seen as caused by dynamic changes in friction forces. Figures 7–9 indicate that the evolution of the motor power responds to adhesive events but seems to be less indicative than other measurements.

### 3.4. Measurement of Stroke Zero Position

The high-speed analysis of position data shows that adhesive moments in the tribocontact lead to a shift of the position signal around the zero position. The equivalent portion of the position signal is issued per period—i.e., the position of zero position of oscillatory motion (offset). This value increases with increasing disturbances (e.g., adhesive effects) in the tribocontact. Through proper integration of the position signal, the zero-position signal can be extracted and evaluated as tribometric resulting parameter.

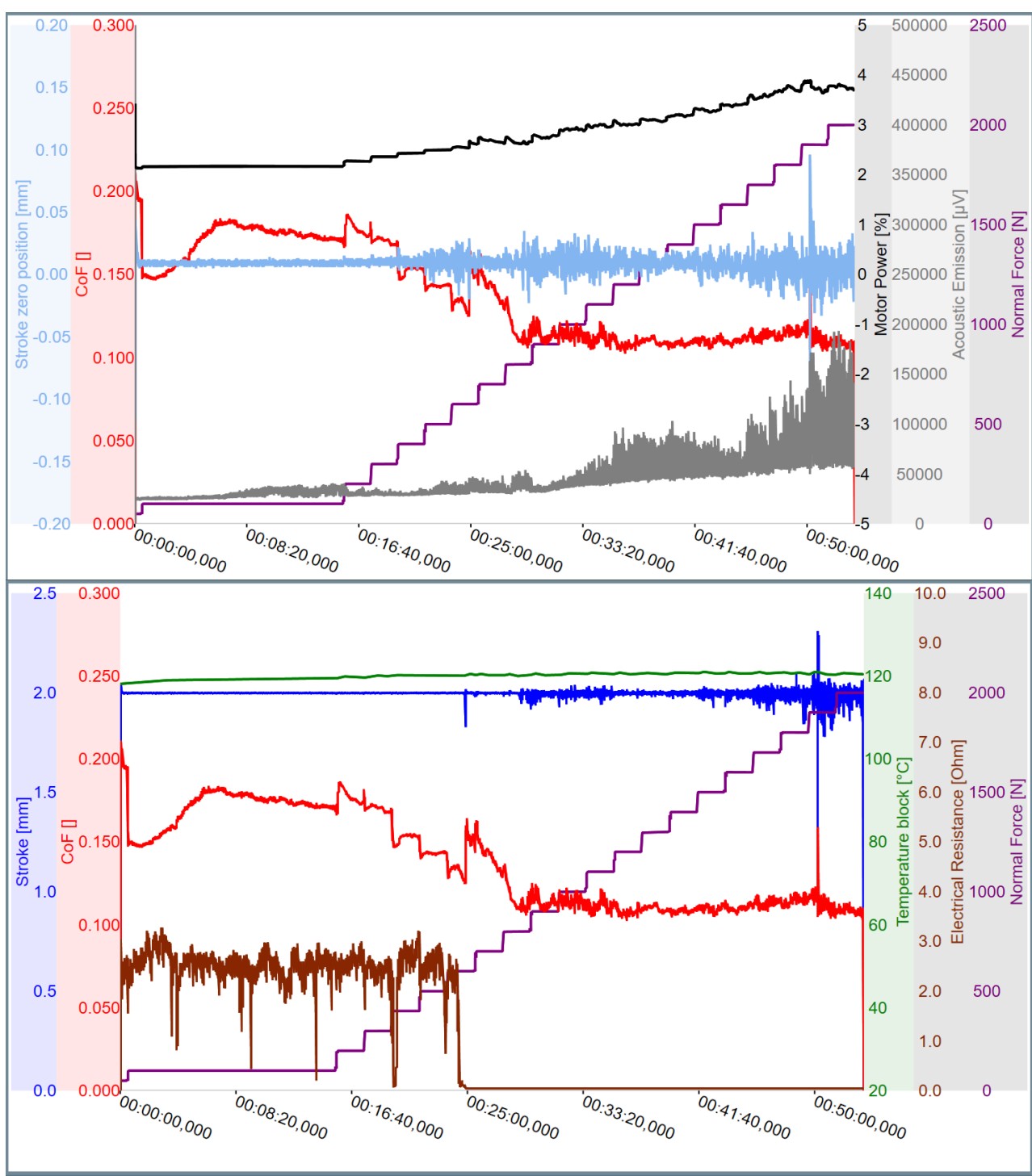

**Figure 8.** Evolution of the coefficient of friction in load step according to ASTM D5706 (top: evolution of the zero-stroke position (light blue), acoustic emission (grey) and motor power (black); lower: evolution of the electrical contact resistance (brown) and stroke (blue)).

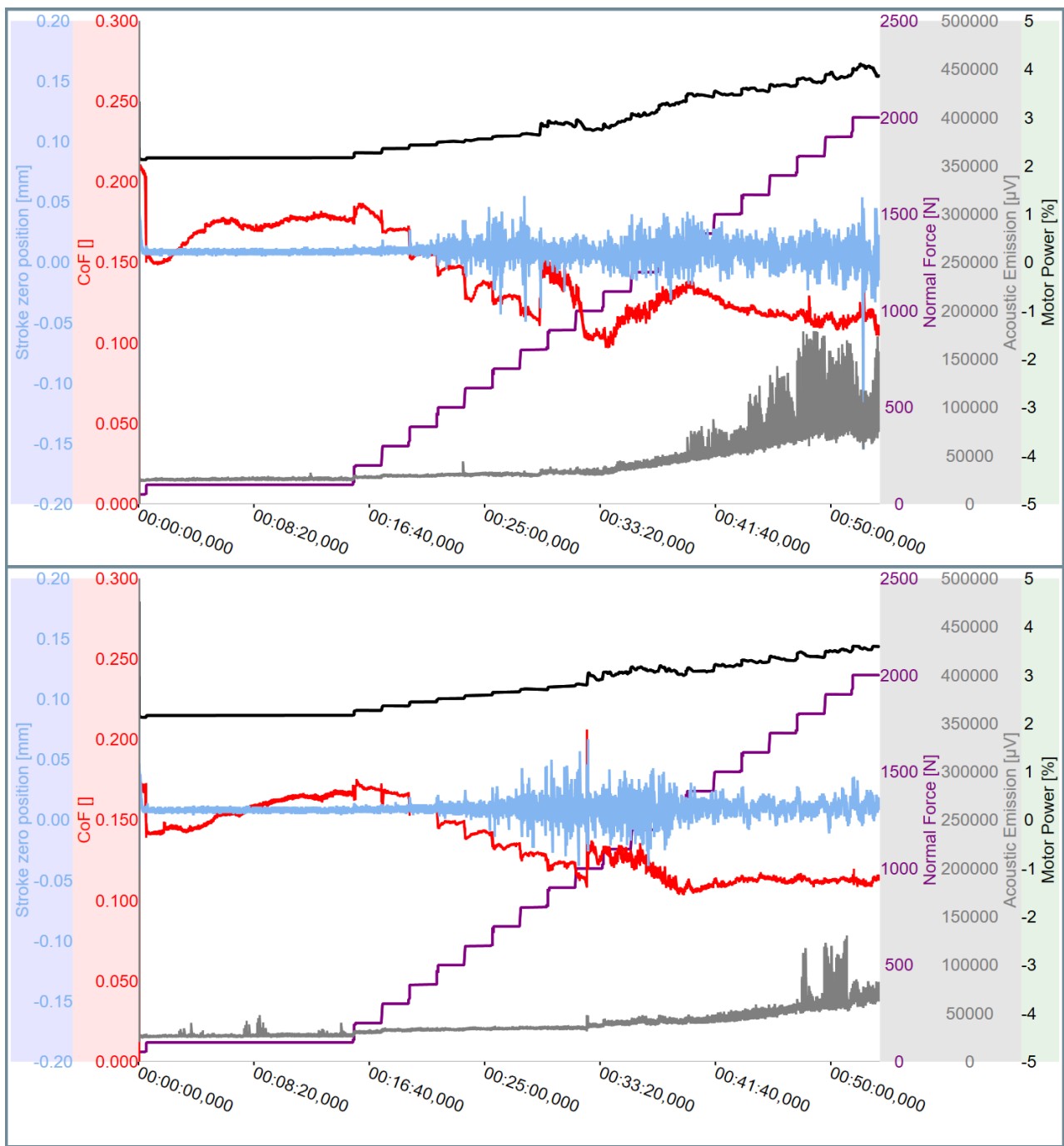

**Figure 9.** Evolution of the coefficients of friction in load step according to ASTM D5706 with additional displays of the zero-stroke position (light blue), acoustic emissions (grey) and motor power (black).

For certain tribosystems, this evaluation shows that the adhesive failure is already announced in advance through increased zero-position shifts (Figures 7–9). For most investigated tribosystems, however, a significant zero-position shift only appears after an initial adhesive event and represents a forward indicator for upcoming changes in the coefficient of friction due to adhesion. A fluctuation of the zero position by more than 0.01 mm can be evaluated as a significant disturbance of the zero position originated by adhesive wear mechanisms. The acoustic emission activities also increased continuously as a consequence of the adhesive wear mechanisms.

The smooth and steady evolution of friction in Figure 10 generated no fluctuations in the evolutions of the additional measurands. The strong peak in the coefficient of friction was, in this example, not noticed in advance by the zero-stroke position or the acoustic emission. The signals of different measurements are ragged only after 300 N, but the fluctuations of the stroke and coefficient of friction were not strong enough to meet the criteria in Table 2. The machine stopped at 1000 N. On the other hand, the sharp drop in the electrical contact resistance indicated a breakdown of the solid tribofilm at 400 N and thus supported the loss of protective functions of the lubricant at that load.

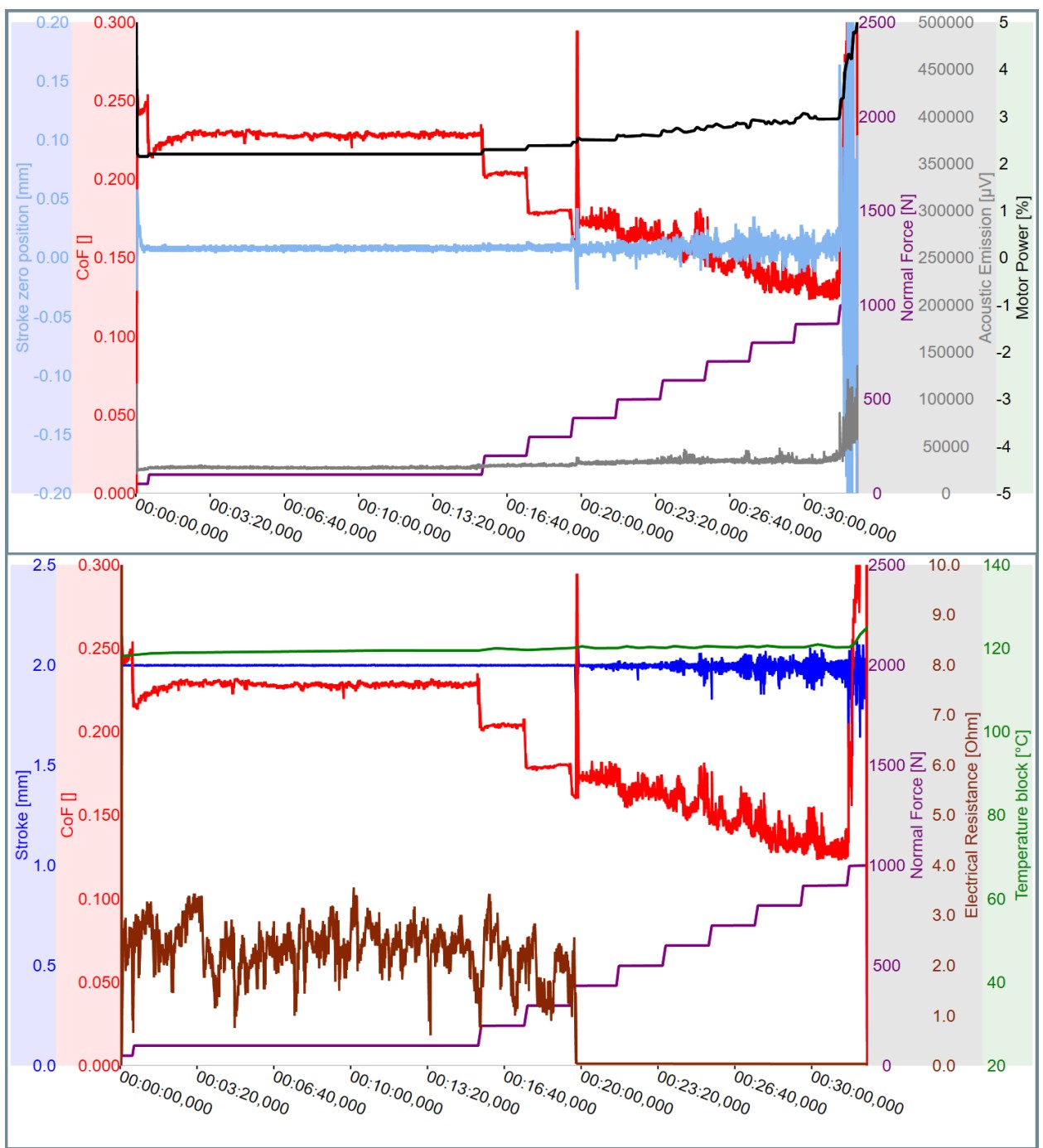

**Figure 10.** Evolution of the coefficients of friction in load step according to ASTM D5706 with an additional display of measurements (top: zero-stroke position (light blue), acoustic emissions (grey) and motor power (black); lower: electrical contact resistance (brown) and stroke (blue)).

## 4. Summary and Conclusions

The new measurements combined with the evolution of the coefficient of friction can be used in different significance as further interpretation tools in seizure load step tests for the identification and/or confirmation of adhesive wear mechanisms. Stroke, stroke zero position, electrical contact resistance and/or acoustic emission go far beyond an evaluation only using the evolution of the friction force signal until the stoppage of the machine. Especially the stroke zero position, electrical contact resistance and acoustic emissions are key indicators which, at an early stage, point towards a loss in the capability to protect the surfaces under extreme pressure conditions.

**Author Contributions:** Conceptualization, G.P. and M.W.; literature review and formal analysis, G.P. and M.W.; writing—original draft preparation, G.P. and M.W.; writing—review and editing, G.P. and M.W.; revisions and supervision, G.P. and M.W. All authors have read and agreed to the published version of the manuscript.

**Funding:** This research received no external funding.

**Institutional Review Board Statement:** Not applicable.

**Informed Consent Statement:** Not applicable.

**Data Availability Statement:** Not applicable.

**Conflicts of Interest:** The authors declare no conflict of interest.

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
