# Peer review of "New Methodologies Indicating Adhesive Wear in Load Step Tests on the Translatory Oscillation Tribometer"

_lubricants, doi:10.3390/lubricants9100101_

Round 1
Reviewer 1 Report
Overall, nice paper!
Comments:
Abstract:
- Perhaps, give explanation of abbreviation of SRV
Abstract, introduction and summary
- It does not become clear, for which application areas the testing procedure and its further development is most relevant
- Perhaps give some application examples, for which the SRV is used to determine adhesive wear protection of lubricants
Figure 2:
- Numbering of components in photograph would be helpful (or leader lines)
Chapter 3.2
- Typo in first sentence „[…] through the specimen.“
Author Response
Replies are marked in blue.
The criticism on English shall be solved and the professional translator didn´t share this criticism.
Reviewer #1
Overall, nice paper!
Comments:
Abstract:
- Perhaps, give explanation of abbreviation of SRV. (see introduction).
Abstract, introduction and summary
- It does not become clear, for which application areas the testing procedure and its further development is most relevant. As scuffing, galling, scoring or adhesive wear occurs everywhere, an allocation to specific applications makes no sense. Second, the manuscript clearly deals with tribotesting of the extreme pressure properties of greases and oils.
- Perhaps give some application examples, for which the SRV is used to determine adhesive wear protection of lubricants. See above.
Figure 2:
- Numbering of components in photograph would be helpful (or leader lines). Done.
Chapter 3.2
- Typo in first sentence „[…] through the specimen.“ corrected.
Reviewer #2
- The abstract should be revised. The abstract was shortened and densified. The significances in engineering field should be highlighted. See introduction and reply to reviewer #1.
- The authors are suggested to explain the novelty of the paper. That´s why only few references are available.
- The authors are suggested to add more references related to the paper. We gladly add related references, if suggested.
- For Figure 1 and Figure 2, the authors are suggested to redraw it. It is not very clear. This remark is unclear for Figure 1. Figure 2 was relabelled according to reviewer #1.
- For the experimental setup, the schematic diagram of the experimental apparatus should be added in the paper. We think, that a schematic illustration has no more details than the actual photo in figure 2, which was relabeled.
- For the lubrication regime in Section 3.1, further step lubrication regimes analysis should be added. It is well known, that extreme pressure tests of lubricants run under the regime of mixed/boundary lubrication. The authors are suggested to add comments as well as the references below. These references clearly illustrate the lubrication regimes as well as the surface roughness effects on the interface. The standard tests used smoothly finished specimen with a predefined topography, which remained logically unchanged.
[1] Theoretical and experimental research on the micro interface lubrication regime of water lubricated bearing. Mechanical Systems and Signal Processing, 2021, 151: 107422. This paper deals with the impact of surface topography and cavities on design and performance of water lubricated journal bearings. What would be the link to adhesive wear/failure and tribometry?
[2] Numerical analysis of added mass and damping of elastic hydrofoils[J]. Journal of Hydrodynamics, 2020, 32(5): 1009-1023. Any connections between this paper and adhesive wear remained invisible from reading the abstract.
[3] An investigation on the lubrication characteristics of floating ring bearing with consideration of multi-coupling factors. Mechanical Systems and Signal Processing, 2022, 162: 108086. This work illuminates design criteria for floating ring bearings. What would be the link to adhesive wear/failure and tribometry?
- For the conclusions, it is too long. Are seven lines too long? Several brief points of conclusions are enough. The authors are suggested to rewrite the conclusions.
- The English should be improved considerately.
Reviewer #3
- The abstract should be rewritten to highlight the study methods and results. The abstract was shortened and densified.
- Please decrease the number of keywords. Done
- The introduction should be rewritten to highlight the studies have been done in the related areas and more literatures should be added. We will be happy to include more references, which combine friction with other quantities and are open for suggestions.
- The number of each part should be added in Fig. 2. Done, see reviewer #1.
- The test condition (load, velocity, temperature, et al.), material, lubrication et al. should be given. First, Table 1 displays test conditions. Second, further details on materials etc. are not necessary, because the contents of the whole paper refer to ASTM D5706 or D7421. Third, this paper deals with tribometric issues and not with the performance assessment of different lubricants. In this respect is the disclosure of properties of the tested lubricants irrelevant.
- The writing format of the paper should be improved. ???

Reviewer 2 Report
Review Comments:
- The abstract should be revised. The significances in engineering field should be highlighted.
- The authors are suggested to explain the novelty of the paper.
- The authors are suggested to add more references related to the paper.
- For Figure 1 and Figure 2, the authors are suggested to redraw it. It is not very clear.
- For the experimental setup, the schematic diagram of the experimental apparatus should be added in the paper.
- For the lubrication regime in Section 3.1, further step lubrication regimes analysis should be added. The authors are suggested to add comments as well as the references below. These references clearly illustrate the lubrication regimes as well as the surface roughness effects on the interface.
[1] Theoretical and experimental research on the micro interface lubrication regime of water lubricated bearing. Mechanical Systems and Signal Processing, 2021, 151: 107422.
[2] Numerical analysis of added mass and damping of elastic hydrofoils[J]. Journal of Hydrodynamics, 2020, 32(5): 1009-1023.
[3] An investigation on the lubrication characteristics of floating ring bearing with consideration of multi-coupling factors. Mechanical Systems and Signal Processing, 2022, 162: 108086.
- For the conclusions, it is too long. Several brief points of conclusions are enough. The authors are suggested to rewrite the conclusions.
- The English should be improved considerately.
Major revision.
Author Response

(The authors gave the same response as above.)

Reviewer 3 Report
- The abstract should be rewritten to highlight the study methods and results.
- Please decrease the number of keywords.
- The introduction should be rewritten to highlight the studies have been done in the related areas and more literatures should be added.
- The number of each part should be added in Fig. 2.
- The test condition (load, velocity, temperature, et al.), material, lubrication et al. should be given.
- The writing format of the paper should be improved.
Author Response

(The authors gave the same response as above.)

Round 2
Reviewer 2 Report
The paper can be accepted for publication now.Reviewer 3 Report
I agree to publish this paper in this journal